# FAST TWO-PHOTON MICROSCOPY BY NEUROIMAGING WITH OBLONG RANDOM ACQUISITION (NORA)

## ABSTRACT

Advances in neural imaging have enabled neuroscientists to study how the activity of large neural populations produce perception, behavior and cognition. Despite many developments in optical methods, there exists a fundamental trade-off between imaging speed, field of view, and resolution that limits the scope of neural imaging, especially for the raster-scanning multi-photon imaging needed for imaging deeper into the brain. One approach to overcoming this trade-off is computational imaging, in which an imaging system efficiently encodes the target image through its optical design and then recovers the acquired information through inverting the encoded measurements algorithmically. Computational imaging thus fundamentally depends on the reliability of recovery. While such approaches are emerging for recovery of optical neural imaging from encoded measurements, they lack a core theoretical sampling theory that will guarantee reliable and accurate recovery. We present here such a theory, based on the widely used model of functional optical imaging videos being low-rank. We show that under simple blurring and randomized line-subsampling conditions, full videos can be recovered from a small fraction of the lines, providing the opportunity for an order-of-magnitude speedup. We use this theory to develop a practical design for fast imaging: Neuroimaging with Oblong Random Acquisition (NORA). NORA, guided by our theory, can be implemented through simple-to-implement changes to widely available systems. Moreover, following our theory, NORA reconstructs the entire video together via nuclear-norm minimization on the pixels-by-time matrix, rather than more common frame-by-frame recovery. We simulated NORA imaging using the Neural Anatomy and Optical Microscopy (NAOMi) biophysical simulator, showing that NORA can accurately recover 400 $\mu$m X 400 $\mu$m fields of view at subsampling rates up to 20X despite realistic noise and motion conditions, thereby demonstrating that our theory holds. These speeds open up the capability of future systems to extend into imaging faster processes in neural systems, such as voltage and glutamate.

## 1 INTRODUCTION

Discovering principles of neural computation at the cellular level rests on the ability to record large populations of single cell activity. Two-photon microscopy (TPM) enables the imaging of activity at a large scale deep within the tissue while maintaining high spatial resolution, resulting in its widespread use in neuroscience. For standard TPM microscopes, the effective speed of such standard systems enable scanning of reasonably large field of views (FOV) on the order of 0.5-1.0 mm square FOVs at approximately 30 Hz. However, while $\approx$30Hz is sufficient for capturing dynamics from calcium indicators, the frame-rate is insufficient for capturing faster neural dynamics, e.g., from voltage indicators that require at a minimum $\approx$10X faster frame rate to effectively record.

TPM is restricted in its frame-rate to its raster-scanning nature, in which the FOV is acquired by scanning point-by-point. As a result, TPM faces a fundamental trade-off between temporal and spatial resolution. Specifically, improving the frame rate requires reducing the amount of samples acquired per frame resulting in poorer spatial resolution. Conversely, fully sampling a FOV lowers the frame rate.

Addressing this speed-resolution-area tradeoff is incredibly important when seeking to image fast processes at high resolution across large FOVs. Thus numerous approaches for high-speed, large-scale imaging Wu et al. (2021); Lu et al. (2017); Song et al. (2017); Demas et al. (2021); Mattison et al. (2023); Kazemipour et al. (2019) have been sought. Most approaches develop purely optical designs that spread light across the tissue Lu et al. (2017); Song et al. (2017); Demas et al. (2021). Examples include volumetric imaging via bessel beams Lu et al. (2017) or stereoscopic imaging Song et al. (2017), multiplane imaging through various forms of beam-splitting Demas et al. (2021), and ROI-specific sampling Mattison et al. (2023). Computational imaging methods, which utilize the co-design of hardware and software, have become another way to address the tradeoff between frame rate and spatial resolution Kazemipour et al. (2019).

While effective, these methods all require collecting light from each fluorescing object in the FOV. Given that the power constraints on *in vivo* imaging systems and the scattering nature of the tissue, spreading the light too much will quickly reduce signal-to-noise levels beyond recoverability and thus effectively cap the maximum effective improvement of such approaches. Furthermore, many of these methods are vulnerable to motion artifacts due to the need for pre-established structural priors, restricting the generalizability of such systems. Finally, many of these methods lack theoretical guidance, requiring a combination of intuition and experimentation to determine reasonable operating procedures and preventing more extreme encoding and sub-sampling that can further improve frame-rates.

Here we start by analyzing a much simpler approach of "blurring and subsampling": two simple and easily implemented optical operations. We consider a scheme where we sample only a fraction of the lines in each frame, mitigating information loss by a small (3-5 $\mu$m) blur in the slow-scan direction, effectively summing 3-5 lines into a single set of measurements. For this sampling scheme, we prove a theorem that shows that under mild assumptions on the rank of the recorded video (a common assumption used widely in denoising and segmenting TPM videos), we can pose video recovery as a matrix completion problem Candes & Plan (2010) and prove reconstruction recovery guarantees.

Using this theory, we develop an ultra-fast TPM microscope that overcomes the speed-resolution-area tradeoff. Our method, Neuroimaging by Oblong Random Acquisition (NORA), only requires simple hardware and well-studied optimization-based reconstruction methods. Crucially, instead of considering each frame of measurement in isolation, NORA considers the measurements as a whole across both its spatial and temporal components. Throughout the entire pipeline, there is no need for complicated optical components or *a-priori* information about the imaging target nor extensive computational resources.

We demonstrate the capabilities of the NORA pipeline by utilizing an advanced two-photon microscopy simulator Song et al. (2021). We show that with NORA, it is possible to achieve 2P recordings with up to 1/20th of the original raster lines even with realistic obstacles such as noise and motion. From the combined design of simple optical changes and holistic data recovery, NORA address a significant bottleneck in increasing the frame rate of two-photon microscopes, enabling possible applications with larger FOVs and faster indicators.

## 2 BACKGROUND

**Multi-photon imaging:** Multi-photon imaging leverages the nonlinear nature of photon absorption to improve optical resolution deeper into scattering tissue. Multi-photon excitation provides highly-localized excitation, which enables light collection from more specific locations of the tissue Luu et al.. Combined with fast galvanometer raster scanning technologies, this excitation "point" can be swept across the tissue to sample across a specified FOV. Raster-scanning is a primary factor behind 2P imaging's high spatial resolution, but the time needed to raster-scan the full FOV also poses a frame rate bottleneck, limiting its use in capturing ultra-fast neural dynamics. A straightforward way to circumvent this bottleneck would be to reduce the number of scanning lines, effectively increasing the frame rate. However, restricting the raster-scanning sacrifices spatial resolution for temporal resolution. Addressing the spatial-temporal resolution tradeoff is thus a major motivation for novel multi-photon imaging systems.

**Computational imaging**: Computational imaging is an approach for co-designing a combination of novel optics with post-processing algorithms by which the optics encodes the incoming light and

the algorithm decodes the measured, encoded samples into a reconstructed image. Computational imaging relies on a modeling the optics encoder through a *forward model*

$$\boldsymbol{y} = g(\boldsymbol{x}) + \boldsymbol{\epsilon}, \tag{1}$$

that describes how the $N$-pixel image $\boldsymbol{x} \in \mathbb{N}$ is encoded into $M$ samples $\boldsymbol{y} \in \mathbb{R}^M$, where here $\boldsymbol{\epsilon} \in \mathbb{R}^M$ represents measurement noise. $g(\boldsymbol{x})$ can be, for example, the mixing of light through a scattering medium (e.g., the diffuser-cam Antipa et al.), a stereoscopic projection Song et al. (2017), or a tomographic projection Kazemipour et al. (2019). Recovering the image $\boldsymbol{x}$ from the coded measurements $\boldsymbol{y}$ is accomplished through an *inverse problem* that finds that image that best explains the measurements through the forward model

$$\widehat{\boldsymbol{x}} = \arg\min_x \left\| \boldsymbol{y} - g(\boldsymbol{x}) \right\|_2^2 + \mathcal{R}(\boldsymbol{x}), \tag{2}$$

where the first term penalizes mismatches to the data and $\mathcal{R}(\boldsymbol{x})$ is a regularization term that promotes known statistics of the image $\boldsymbol{x}$. For example, $\mathcal{R}(\boldsymbol{x})$ can enforce smoothness though the so-called total variation norm Rudin et al.. Regularization becomes especially important when the optics forward model is not well posed, i.e., many possible images $\boldsymbol{x}_k$, for $k = 1, ..., K$ can re-create the same data equally well: $\left\| \boldsymbol{y} - g(\boldsymbol{x}_{k_1}) \right\|_2^2 \approx \left\| \boldsymbol{y} - g(\boldsymbol{x}_{k_2}) \right\|_2^2$. The regularization serves as a "tie-breaker", making the problem well posed again by further restricting the solution to best align with the expected statistics of the target data.

**Compressive sensing and matrix completion**: Compressive (or compressed) sensing (CS) Romberg (2008) and matrix completion Candes & Plan (2010) recover signals or data from limited measurements by leveraging underlying structure (sparsity or low rank). In compressive sensing, we recover a $K$-sparse vector $\boldsymbol{x} \in \mathbb{R}^N$, i.e, $K \ll N$ elements of $\boldsymbol{x}$ are non-zero, from undersampled linear measurements $\boldsymbol{y} = \boldsymbol{A}\boldsymbol{x} + \epsilon$. Here $\mathbf{y} \in \mathbb{R}^M$ for $M < N$. Classical CS theory guarantees the recovery of $\boldsymbol{x}$ from $\boldsymbol{y}$ when, e.g., $\boldsymbol{A}$ satisfies the Restricted Isometry Property (RIP) Candes (2008) or Null-Space Property (NSP) Mansour & Saab (2017) and produces enough measurements, typically of the form $M \geq CK \log(N)$ for a constant $C$, i.e., $\boldsymbol{y}$ can be much smaller than $\boldsymbol{x}$ and still recover it accurately. Often the properties over $\boldsymbol{A}$ not hold absolutely, but with high probability if $\boldsymbol{A}$ is a random sensing matrix. While CS works well for certain distributions over $\boldsymbol{A}$, e.g., *i.i.d.* Gaussian or random Fourier, it does not work well for others, e.g., 1/0 random Bernoulli elements.

Similar to CS, matrix completion recovers a low-rank ($R$) matrix $\boldsymbol{X} \in \mathbb{R}^{N \times T}$ from a subset of its entries, leveraging the low-rank property instead of sparsity. $\boldsymbol{X}$ can be recovered via nuclear norm minimization, a convex relaxation of rank minimization:

$$\widehat{\boldsymbol{X}} = \arg\min_{\boldsymbol{X}} \|\mathcal{A}(\boldsymbol{X}) - \boldsymbol{y}\|_2^2 + \lambda \|\boldsymbol{X}\|_*, \tag{3}$$

where $\boldsymbol{y}$ are the observed entries and $\mathcal{A}(\boldsymbol{X})$ selects the corresponding entries of $\boldsymbol{X}$. For matrix completion, similar recovery conditions guarantee accurate estimation of $\boldsymbol{X}$

$$M \geq CR\left(N + T\right)\log\left(NT\right). \tag{4}$$

Here the number of samples $M$ required to recover $\boldsymbol{X}$ depends on the latent dimensionality $R(N + T)$, rather than the full number of samples $NT$, leading to a much higher sampling efficiency Candes (2010); Candes & Plan (2010).

## 3 NORA SAMPLING AND RECOVERY

At a high level, our approach seeks to minimize the acquisition time per frame by reducing the required number of line-scans, a major reason behind the framerate bottleneck in TPM. We achieve this reduction through random subsampling and blurring, each of which are simple to implement physically as well as being well-modeled by a linear operator. The recovery approach is based on matrix-completion, an estimation approach that can leverage space-time correlations (Fig. 1). This framework enables a theoretical understanding of the expected reconstruction accuracy due to its well-defined linear model and the use of low-rank priors, which provide provable guarantees for recovery under subsampling conditions. The combination of simple optical front-end and algorithmic

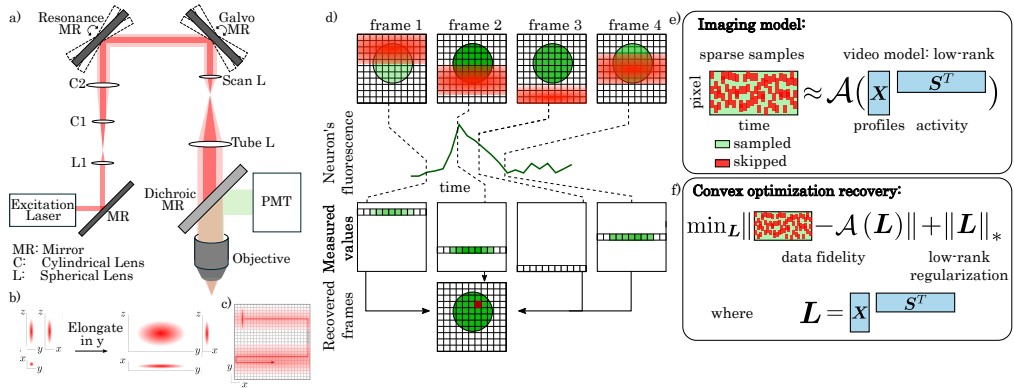

Figure 1: Overview of NORA. a) blurring and subsampling can be achieved in a the standard two-photon microscope with the addition of two cylindrical lenses to achieve an elongated PSF. b) Elongated PSF shape used in NORA sampling. c) Example sampling pattern depicting the blur-and-subsample approach. d) Schematic depicting the recovery process. A single pixel is integrated in multiple wide-line scans. The combined information over multiple frames, along with the correlations between frames, provides enough information to triangulate the value of the single high-resolution pixel at all frames. e) The imaging model quantifies the optical path as a blur operator applied to a low-rank fluorescence video matrix, followed by a sub-sampling operator. Importantly, all operations in the forward model are linear. f) The recovery process solves a nuclear-norm regularized least-squares optimization. This optimization finds the video that both matches the observed samples and the *a priori* model of fluorescence video being low-rank.

back-end enables an efficient and reliable imaging system capable of accurate reconstruction even with extreme subsampling.

**Matrix-completion based recovery theory:** To develop a theory, we focus on two basic operations that lend themselves well to TPM: subsampling and blurring. Mathematically, we denote $\boldsymbol{X} \in \mathbb{R}^{N \times T}$ to represent our original imaging target across time, where $N$ is the number of pixels in each frame and $T$ is the total number of frames. We write the forward model as

$$\boldsymbol{Y} = \mathcal{A}(\boldsymbol{X}) + \boldsymbol{E} = \boldsymbol{S}(\boldsymbol{B}\boldsymbol{X}) + \boldsymbol{E}, \tag{5}$$

where we denote the blur operator as $\boldsymbol{B}$[1] and the linear sampling operator as $\boldsymbol{S}(\cdot)$. The linear sampling operator specifically returns for the $k^{th}$ column (i.e, the blurred $k^{th}$ frame) the samples (lines) taken for that frame. Note that $\boldsymbol{S}(\cdot)$ samples each frame with a different pattern for every frame. Thus, while $\boldsymbol{S}(\cdot)$ is still linear, it cannot be written as a single matrix that can be applied to all frames, as $\boldsymbol{B}$ can. $\boldsymbol{Y} \in \mathbb{R}^{M \times T}$ denotes our observed measurements, where each column is the set of $M$ blurry measurements selected at that frame. $\boldsymbol{E}$ is the $M \times T$ measurement error matrix.

With this measurement acquisition scheme, we next focus on recovering the imaging target, i.e., the full, non-blurry video, by inverting the forward model. However, this inversion is a non-trivial problem. Despite the model being linear, using a direct pseudo-inverse is not possible due to the fact that $MT < NT$. The under-determined nature of the inverse problem means that an infinite number of solutions exist. To compensate for the incomplete measurements, we model our data as low-rank, a common model in TPM denoising and segmetation that takes advantage of local spatial and temporal redundancy resulting in intrinsically low-dimensional fluorescence microscopy data.

We frame, as in much of the computational imaging and inverse problem literature, the high-level goal of determining an estimate $\widehat{\boldsymbol{X}}$ that satisfies both our observed measurements and our *a priori* model of low-rank fluorescence video. Imposing this a-priori constraint enables us, as it has had in many other applications, the recovery of $\boldsymbol{X}$ despite the ill-conditioned form of the forward model. Mathematically we find the estimate $\widehat{\boldsymbol{X}}$ through the nuclear-norm formulation of low-rank matrix completion (Fig. 1f), i.e., we optimize the cost function

$$\widehat{\boldsymbol{X}} = \arg \min_{X} \left\| \boldsymbol{Y} - \boldsymbol{S}(\boldsymbol{B}\boldsymbol{X}) \right\|_F^2 + \lambda \|\boldsymbol{X}\|_*, \tag{6}$$

---

[1]We note that $\boldsymbol{B}$ has a complex 2-level block circulant structure, however the form of $\boldsymbol{B}$ is not important as all that matters is that we can apply the blurring operation to any image. This application can be done in the image domain as a convolution, and the notation here is simply for mathematical efficiency.

where the first term (Frobenius norm) is the data fidelity term that enforces the estimate to match the observed samples, and the second term is the nuclear norm (defined as the sum of singular values) that prioritizes low-rank solutions. $\lambda$ is a parameter that trades off between these terms. Solving this optimization has been well studied in the literature. For our implementation, we selected an efficient first-order optimizer that is readily available (Becker et al., 2011).

The nature of nuclear norm optimization for linear forward models permits theoretical understanding of the expected reconstruction accuracy which scales linearly with $\min(T, N)$. Specifically, for blur-and-subsample imaging of video sequences we can derive the following guarantees:

**Theorem 1.** *Assume $L'$ samples are taken at every frame for $T$ frames, i.e., the total number of measurements is $M = TL'$. Assume $TN \geq M$, $N > R$, and $N, T > O(1)$. When*

$$M \geq C\beta R(T\mu_b^2 + N)\log^2(NT).$$

*Then with probability at least $1 - O\left((TN)^{-\beta}\right)$ the solution to Equation equation 6 produces an estimate $\widehat{\boldsymbol{X}}$ of the rank-$R$ matrix $\boldsymbol{X}$ with an estimation accuracy bounded by*

$$\left\|\boldsymbol{X} - \widehat{\boldsymbol{X}}\right\|_F \leq \left(4\sqrt{\min(T, N)\left(2NT + M\right)/M}\right)\epsilon,$$

*where $\epsilon$ is the per-pixel noise error and $\mu_b^2$ measures the coherence of the left singular vectors of $\boldsymbol{X}$ and the PSF $\boldsymbol{b}_n$ at different locations $n$: $\mu_b^2 = (N/R)\max_n \|\boldsymbol{U}^T\boldsymbol{b}_n\|_2^2$.*

Theorem 1 demonstrates that when data is intrinsically low-rank, the sampling rate required for accurate reconstruction can be quite low, i.e., proportional to the rank times the sum of the matrix dimensions, rather than the full number of elements in the video matrix. (see Supplement for the full proof).

**Implementation motivated by theory:** Our theory indicates that simple optical operations, combined with image recovery, has the potential to vastly improve imaging frame-rates. Building on this theory, we develop NORA: Neuroimaging with Oblong Random Acquisition (Fig. 1.a).

NORA implements sub-sampling via random line-scanning, which can be achieved by controlling the scanning pattern of the scanning mirrors in a TPM system. In a standard TPM system, the FOV is sampled through the combination of two scanning mirrors: one fast scanning mirror moves the excitation beam in a line, while the other, slower, scanning mirror moves the beam to its next line position. In NORA, we modify the slow-scan mirror to purposefully skip lines, randomly varying the distances between each line scan. This random subsampling decreases the number of line-scans taken per frame while still diversifying the information captured between concurrent frames by avoiding imaging the same locations. Random line-scanning is critical to enabling the imaging speedup due to the proportional relationship between number of scan-lines and per-frame acquisition time. While this work simulates reconstruction for straight line-scans, the NORA method easily extends to angled line scans, which can be more naturally achieved resonance scanner-based systems by smoothly modulating the galvameter speed. All that is required is that the diversity of measurement locations is maintained and the sample location is known.

When the number of sampled lines become increasingly small, a simple subsampling-only system could pose a significant limit on the total amount of information that could be acquired. To overcome this limitation, in addition to the subsampling, NORA implements blurring via an excitation beam that is elongated in the slow-scan direction, resulting in an oblong point spread function (PSF). This elongation is optically simple and inexpensive to implement, only requiring a pair of cylindrical lenses in the beam conditioning path to introduce the necessary astigmatism Li et al. (2024). An oblong PSF mitigates information loss along the slow-scan direction by effectively summing together neighboring scan lines, allowing fewer raster scans lines to be required per frame to capture information from our target. Elongating the PSF solely along the slow scan direction also ensures that the other axis, the fast scan direction, maintains its original resolution. Furthermore, it is not so elongated that it covers the area of all of the unsampled scan lines (i.e., it is not a full "line scan" as in Kazemipour et al. (2019)). Rather, the goal is to simply ameliorate some of the lost information. The resulting subsampled-and-blurry measurements will still be missing information from the target, but only if each frame is seen in isolation.

The power of NORA's recovery algorithm lies in its use of information shared *between* frames, as per our theory that focuses on simultaneous multi-frame recovery. $\boldsymbol{S}_t$ is time-dependent, resulting

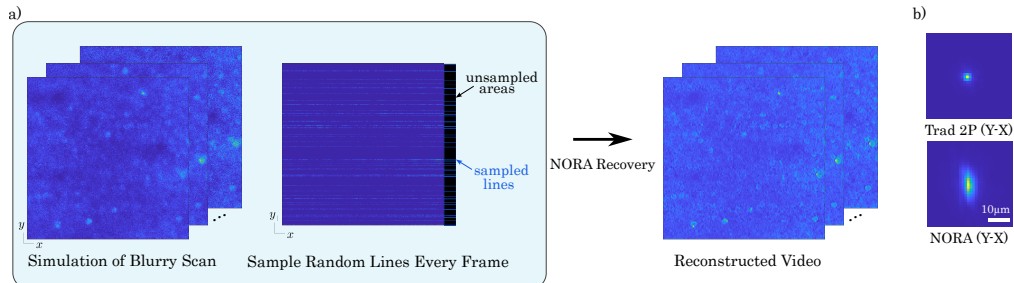

Figure 2: NAOMi simulation of NORA imaging. a) NAOMi simulates a full 3D volume of tissue *in silico*, which is then imaged with the desired PSF through an optics model that includes tissue-based aberration and photonic variability. The result is a blurry image of the tissue at each frame. Biophysical models of neural activity ensure realistic changes in fluorescence for the simulated neurons and processes, as well as for neuropil. Frame-specific sampling then selects the per-frame lines to keep, which are then inverted through the matrix completion recovery to obtain an estimate of the videos. Simulations through NAOMi enable the recovered video to be compared to "standard" imaging obtained through simulating a video sequence on the same volume and activity, but with a typical PSF and no subsampling. b) Example PSF used in the NAOMi simulation (bottom) compared to a traditional 2P PSF (top).

in different randomly selected lines at different frames. Thus areas not captured on a frame at time $t$ are likely to be captured at frames in time $t-1$, $t+1$, etc. It is then up to the reconstruction algorithm to leverage these diverse measurements across multiple frames to not just deblur the lines that were collected with the spread PSF, but to fill in the missing areas by utilizing a low-rank prior. The ability to acquire signals from multiple lines simultaneously through the oblong PSF allows for a further reduction in the number of raster lines required per frame while still effectively capturing that frame's information from our target.

Our easy-to-implement design eliminates many of the difficulties associated with high-speed two-photon microscopy such as cost and expertise. Both the data acquisition and reconstruction aspects of NORA solely rely on simple changes to pre-existing structures. The hardware design of the NORA system is remarkably similar to the traditional two-photon system. Likewise, the reconstruction does not rely on algorithmically complicated and expensive steps such as model pre-training or gathering large training datasets. These two components taken together removes major barriers towards ultra-fast microscopy.

## 4   RESULTS

**Validation through Simulation:** We validate our design through simulation of our NORA optical path in a biophysical simulation suite followed by reconstruction using the matrix completion approach (Fig. 2). We simulated the subsampled-and-blurry measurements through the NAOMi simulator (MIT license), a MATLAB-based two-photon microscopy simulator that can create highly-realistic movies of neural tissue with full ground truth Song et al. (2021). It is important to emphasize that in our simulation, we do not simply take pre-recorded videos and apply the forward model $S(B)$. Rather, NAOMi simulates the full 3D point-spread function and imaging pathway, inclusive of motion, sensor-realistic noise, etc. Using these realistic simulations, we generated a 400 $\mu$m by 400 $\mu$m by 100 $\mu$m volume of tissue mimicking the anatomy of mouse layer 2/3 in area V1 at 250 $\mu$m depth with GCaMP6f. The volume was fully scanned with an elongated PSF (FWHM of 1.15 $\mu m$ by 3.35 $\mu m$ by 6.99 $\mu m$) at 120 mW power and 30 Hz frame rate over 1000 frames. The resulting blurry video was then subsampled by taking random columns form each frame.

We simulated scans with realistic nuisance variation, including noise and motion, as validated in the original NAOMi paper. We tested different levels of subsampling by undersampling the same video at 10X, 15X and 20X (i.e., 1/10th, 1/15th, and 1/20th of the total lines, respectively). Moreover, while we included tissue motion to simulate the most realistic settings, we further tested the impact of motion by comparing the results to the same video generated with the motion "turned off".

We developed the reconstruction algorithm in MATLAB[2], using the the Templates for First order Conical Solvers (TFOCS) library (BSD 3-clause license) to solve a modified matrix completion

---

[2]Code for both simulation and reconstruction will be released with the publication of this paper.

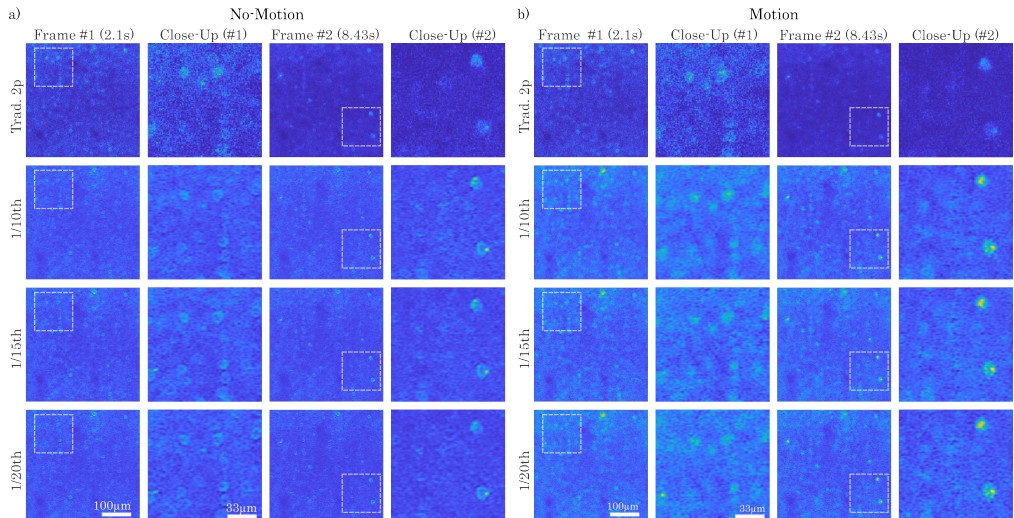

Figure 3: Spatial recovery results for NORA imaging using NAOMi simulations with no motion (a) and with motion (b). (a) and (b) both show example single frames and insets from using (from top to bottom) a traditional 2P PSF, NORA imaging at 10X, 15X, and 20X speedups ($1/10^{th}$, $1/15^{th}$, and $1/20^{th}$ of the lines per frame, respectivly). While (a) maintains a slightly higher level of spatial resolution, individual cells are still clearly identifiable up to NORA at 20X for both cases.

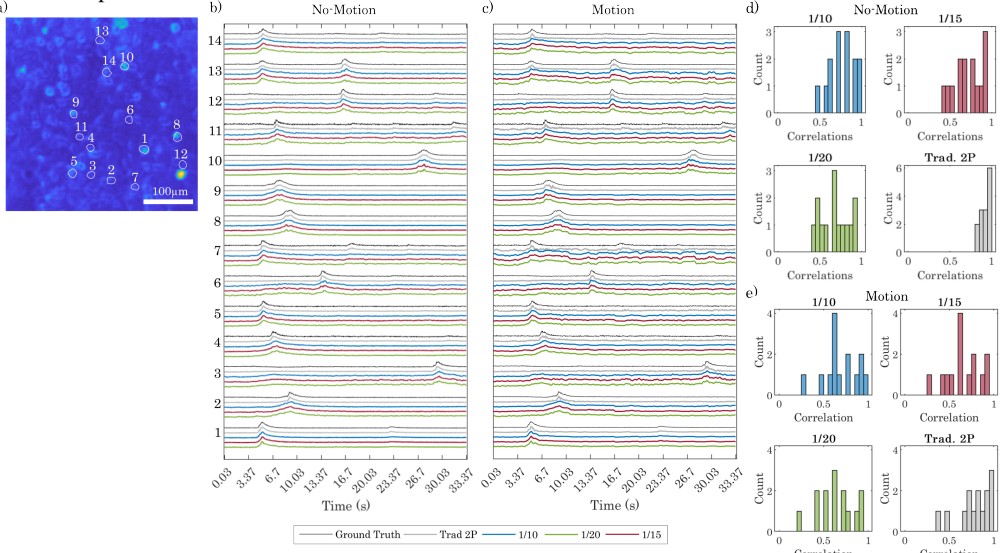

Figure 4: Temporal activity recovery results for NORA imaging. a) Standard deviation image of the median filtered reconstruction with circled ROIs representing individual neurons with significant activity, as identified from NAOMi simulator. b) Time traces from for the ROIs in (a) for simulations without motion. c) Same as (b) but for simulations with motion. In both (b) and (c), activity is preserved up to 20X speedups. d,e) histograms of correlations between each imaging method and the ground truth time-traces for the cases with and without motion. While traditional 2P imaging has the highest correlations, the drop in correlations are minimal, indicating well-preserved neural activity.

problem Becker et al. (2011). TFOCS offers several advantages, computational efficiency from bring a first-order method and the ability to use implicit functions. These advantages greatly mitigate the computational challenges of recovering entire video sequences. All computational work was performed on an Intel Core i7 8700k CPU desktop with 64GB RAM. Every 500 frames of reconstruction took ≈2 hours.

**Video recovery from NORA samples:** Following the NORA pipeline, we simulated the NORA subsampling and blurring for different subsampling ratios: 1/10th, 1/15th, and 1/20th of the FOV. In Figure 3 we highlight example frames from the NORA-reconstructed videos ($\lambda$ set manually; see Supplement) using frames from a fully-sampled Gaussian PSF scan (traditional 2P) as comparison. We do not include any denoising to highlight the base reconstruction capabilities of NORA.

Despite the extreme subsampling, the NORA reconstructions recover the frames to a visual quality on par with the traditional 2P scan. The reconstruction only shows complete frames of activity without subsampling artifacts and bands of missing information. NORA also remarking maintains a surprising amount of consistency between the reconstructions from different subsampling ratios. As seen through the example frames, there are no major differences between the $1/10^{th}$ subsampling reconstruction and the $1/20^{th}$ subsampling reconstruction.

Motion can increase the rank of a video matrix thus violating the low-rank assumption. Specifically, to test the robustness of NORA to rigid motion, we leveraged the ability of NAOMi to simulate both line-by-line and rigid motion. Despite these concerns, the addition of motion only had a minor affect on the visual quality of the reconstructed videos (Fig. 3a,b). While some details, e.g., reconstruction sharpness, is affected, the relevant information that needed for ROI extraction, e.g., the location and size of each neuron, are successfully recovered in despite motion and high levels of signal recovery are still possible up to 20X speedups. This suggests that although motion does increase the effective rank, the increase is not to the extent that the low-rank model no longer holds.

While the reconstructed videos demonstrated good image recovery, the core information that is critical in neuroimaging applications is the temporal activity of individual ROIs in the FOV. If the neural fluorescence fluctuations are distorted, then the imaging method is not well suited to scientific imaging. Conversely, if the images are noisy but the activity is preserved, the imaging method can be used. To examine the time traces, we compared the ground truth traces from NAOMi with the time-traces extracted from ROIs in the NORA reconstructions. We extracted the ROI traces by masking reconstructed videos with the ground truth spatial profile provided by NAOMi, i.e., the Profile-Assisted Least Squares (PALS) Song et al. (2021) (Fig. 4a), removing possible confounds of algorithmic-specific errors, such as false positive transients Gauthier et al. (2022). To reduce noise, specifically extreme values from the Poisson-Gaussian nature of the PMT noise, and focus on signal recovery, we further median filtered all videos with a small 3D window (9x9x9 pixels) before extracting traces. We further compared the NORA recovered ROI traces to traces from the traditional 2P simulation scan. This comparison provided a reference for how well a typical system would have captured the ground truth activity. Traces are shown for 33.33s of simulated recording (Fig. 4), both for simulations without motion (Fig. 4b,d) and with motion (Fig. 4c).

The traces extracted from the reconstructed volumes are largely in line with the traces from the ground truth for both with and without motion (Fig. 4b,c). Traces such as #1, #3, and #6 show the consistency in the spike between the ground truth and the reconstruction. In cases like trace #7 (Fig. 4c), we see that the traditional 2P traces also fail to match the ground truth, suggesting that the failure is due to a general scanning issue rather than NORA.

One important question we consider is the general trend of the reconstructions: do most traces consistently align with the ground truth? To verify the overall alignment of the extracted traces with the ground truth, the distribution of the correlation between the ground truth traces and the extracted traces are plotted in Figure 4d,e. Even in the simulation condition that includes motion, all traces from the reconstructed volumes show a skew toward the left, indicating a general trend of high correlation with the ground truth despite the inclusion of noise and motion in the measurement process.

## 5 Discussion

In this work we develop theoretical compressive-sensing type bounds on how simple optical setups using blurring and sub-sampling, combined with multi-frame matrix-completion based recovery, can vastly increase imaging speeds in functional raster-scanning two-photon imaging. Our theory is based on the low-rank model of TPM data commonly used in TPM post-processing that captures the high levels of spatial and temporal correlations. Based on this theory we present NORA: a framework that instantiates our theory in a practical approach for fast TPM. The advantages of NORA system are 1) the simplicity of the optical design, 2) the use of both spatial and temporal statistics for video recovery, and 3) theoretical guarantees on recovery. Moreover, NORA uses generic low-rank statistics and thus does not require any training data or model fitting.

One major feature is NORA's robustness to both extreme subsampling and motion. Given that motion increases the effective matrix rank by mis-aligning pixels across frames, we anticipated requir-

ing additional algorithmic improvements to address motion. Despite this rank increase, however, and we were able to accurately reconstruct the video from sub-sampling levels up to a factor of 20X. This successful recovery highlights that NORA is capability of handling realistic data challenges.

In our approach we focused on traditional matrix reconstruction via nuclear norm minimization, which has been highly successful in other domains Candes & Plan (2010); Charles et al. (2017); Ahmed et al. (2013). This approach moves from thinking of compressive microscopy recovery as per-image frame-by-frame recovery to using the full set of spatial and temporal statistics. Moreover, low-rank priors are powerful enough that reconstruction is possible without strong trained priors learned by recent deep learning approaches. We thus bypass the need for training data, and remove the possibility of realistic hallucinations that can be difficult to diagnose. Furthermore, NORA has the capacity to generalize to new tissue samples, unlike deep learning approaches that face significant out-of-distribution challenges Mishne & Charles (2024). Importantly, as a new sampling scheme, there are no prior baselines that tackle the same setting and thus the primary point of comparison is a comparison to traditional imaging.

NORA's simple, practical design can enable many labs to build an ultra-fast microscope with with minimal additional parts and effort. We envision two possible use cases for NORA: fast imaging of a single FOV, and interleaved imaging of multiple FOVs. In the former, a single FOV is sampled as fast as possible, using the full framerate to acquire signals faster than calcium dynamics (e.g., voltage Evans et al. (2023) or glutamate Aggarwal et al. (2023) indicators). This use case has a many applications in studying neural interactions based on fast spike timing. In the latter, random access refocusing can flexibly select multiple (10-20) independent FOVs over the optically accessible volume without specific, required geometrical configurations (e.g., tiling in depth). The second use case opens up the ability to study the network-level activity of multiple brain areas at the single neuron resolution.

**Limitations:** The physical design proposed for NORA only requires simple optical components. One aspect, however, that may differ from simulation is the random sampling implementation. In simulation, each frame's measurements consists of randomly selected columns. A physical system would require the galvanometer to modulate its acceleration to accommodate the varying distances between samples, which could cause problems such as jerk and slow down the system. A preferable sampling would consist of uniform distances between each sample, such as evenly-spaced columns of measurements, to allow for consistent speed between measurement column. The flexibility of NORA allows for this type of sampling, as long as the measurements are sufficiently different across frames. Our work largely focused on the validation of NORA through theoretical guarantees and biophysical simulation. Validation in practice requires extensive simultaneous imaging that, while expensive and time-intensive, represent exciting next steps. Another limitation is the current reconstruction run-time, which we are actively working to improve with better hardware (GPUs) and optimized software.

## ETHICS STATEMENT

The NORA recovery method inherently avoids the risks of hallucinating realistic-but-incorrect reconstructed data that are common in modern ML-based approaches. By relying on low-rank priors rather than learning-based models or pre-trained priors, NORA ensures that reconstructed imaging remains grounded in actual data, even in domains where raw data cannot be visually inspected or ground-truth validation is unavailable. Furthermore, through our use of data simulated through NAOMi, a freely available two-photon microscopy simulator, we are able to validate our performance against a known ground truth, ensuring accessibility and transparency in our results.

## REPRODUCIBILITY STATEMENT

The description of our proposed is provided in Section 3, with implementation details given in Appendix A and the theoretical guarantees shown in Appendix B. The proposed pipeline is simulated through data simulated through NAOMi, a freely available two-photon microscopy simulator which ensures reproducibility of the data shown in this work. The source code for the full pipeline will be released with the publication of this paper.

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

APPENDIX

## A  IMPLEMENTATION DETAILS

Here we provide additional information on parameters necessary to implement the NORA reconstruction. The implementation builds on the templates provided in the TFOCs library Becker et al. (2011), and likewise utilize their formulation of the nuclear norm basis pursuit denoising (BPDN) problem. Overall, the NORA reconstruction is based on the regularized version of the nuclear (BPDN) problem:

$$\widehat{\boldsymbol{X}} = \arg\min_{\boldsymbol{X}} \|\mathcal{A}(\boldsymbol{X}) - \boldsymbol{y}\|_2^2 + \lambda\|\boldsymbol{X}\|_*, \tag{7}$$

where $\boldsymbol{X} \in \mathbb{R}^{N \times T}$ is a low-rank ($R$) matrix that we wish to estimate, $\boldsymbol{y} \in \mathbb{R}^M$ are the observed entries and $\mathcal{A}(\boldsymbol{X})$ is the feed-forward model that describes mathematically how the data in $\boldsymbol{X}$ is projected into our measurements $\boldsymbol{y}$. The choice in value for $\lambda$, the regularizer parameter, determines the relative strength of the low-rank prior.

The TFOCs implementation essentially seeks to solve the same problem but translating the problem into a constrained optimization form. This form further includes a smoothing term in order to make the problem better posed, resulting in the following formulation for Nuclear BPDN:

$$\widehat{\boldsymbol{X}} = \arg\min_{\boldsymbol{X}} \quad \|\boldsymbol{X}\|_* + \frac{1}{2}\mu\|\boldsymbol{X} - \boldsymbol{X}_0\|_F^2 \tag{8}$$

$$\text{subject to} \quad \|\mathcal{A}(\boldsymbol{X}) - \boldsymbol{y}\| \leq \epsilon, \tag{9}$$

where the second term in the cost function is the smoothing term, $\boldsymbol{X}_0 \in \mathbb{R}^{N \times T}$ is the optional initial point, $\mu$ is the smoothing parameter, and $\epsilon$ is the noise parameter. TFOCS can run in 'continuation' model where the solution is fed back into the optimization as the initial point $\boldsymbol{X}_0 \leftarrow \widehat{\boldsymbol{X}}$ until the solution does not deviate much from the initial condition. This allows for some flexibility in the choice of constant $\mu$. For our reconstructions, we performed three iterations with a maximum of 200 inner iterations starting at a $\mu = 0.1$ and decrementing the mu by 0.1 at every outer iteration for all reconstructions.

For our implementation of the NORA reconstruction, careful selection of $\epsilon$ is essential to guaranteeing successful reconstruction. We used the following values of $\epsilon$ for the reconstructions shown in Figures 3 and 4, which were reconstructed in batches of 500 frames:

|  | 1/10th | 1/15th | 1/20th |
|---|---|---|---|
| **Noise, No Motion** | 475 | 375 | 325 |
| **Noise, Motion** | 475 | 375 | 340 |

Table 1: $\epsilon$ values for simulated reconstructions

The above values for $\epsilon$ were chosen manually based on reconstruction quality. We note that there appears to be a relationship between the number of measurements and the value of $\epsilon$, with $\epsilon$ scaling with increased measurements. Intuitively, this could be due to the smaller number of measurements introducing less variations, allowing for a smaller error tolerance. When implementing NORA, we recommend starting with the given $\epsilon$ values, then scaling depending on the number of frames, size of the FOV, and subsampling ratio.

## B  PROOF OF NORA RECONSTRUCTION

This appendix aims to prove Theorem 1, which bounds the recovery error of estimating a low-rank video sequence $\boldsymbol{X} \in \mathbb{R}^{N \times T}$ from the NORA measurements $\boldsymbol{Y} \in \mathbb{R}^M$. The basis of the proof rests on earlier work in the matrix recovery literature that have derived recovery guarantees for similar systems using a dual certificate approach certificate approach Candes (2010); Candes & Plan (2010). In the dual certificate approach, we use the fact that if there exists a "certificate" that satisfies

a series of inequalities derived from the KKT conditions, then the low-rank matrix is recoverable from a nuclear norm minimization.

Specifically, the certificate $\boldsymbol{Z}$ must satisfy two bounds, one on its projections within the span of the singular vectors of $\boldsymbol{X}$, and one on the projection outside of that same span. To prove that such a certificate exists, it is sufficient to find one such instance of a matrix that satisfies all the proper conditions, i.e., "proof by construction".

We start by defining the singular value decomposition of the video sequence $\boldsymbol{X}$ as $\boldsymbol{S} = \boldsymbol{Q}\boldsymbol{\Sigma}\boldsymbol{V}^T$. To be able to simply define the dual certificate properties that must hold, we next define the projections into the space spanned by the left and right singular vectors $\mathcal{P}_T$, and projection into the complement space $\mathcal{P}_{T^\perp}$ as

$$
\begin{aligned}
\mathcal{P}_T(\boldsymbol{W}) &= \boldsymbol{Q}\boldsymbol{Q}^T\boldsymbol{W} + \boldsymbol{W}\boldsymbol{V}\boldsymbol{V}^* - \boldsymbol{Q}\boldsymbol{Q}^T\boldsymbol{W}\boldsymbol{V}\boldsymbol{V}^T \\
\mathcal{P}_{T^\perp}(\boldsymbol{W}) &= (\boldsymbol{I} - \boldsymbol{Q}\boldsymbol{Q}^*)\boldsymbol{W}(\boldsymbol{I} - \boldsymbol{V}\boldsymbol{V}^T).
\end{aligned}
\tag{10}
$$

Given these projection operations, we can state the dual certificate as

$$
\left\| \mathcal{P}_T(\boldsymbol{Z}) - \boldsymbol{Q}\boldsymbol{V}^T \right\|_F \leq \frac{1}{2\sqrt{2}\gamma}
\tag{11}
$$

$$
\left\| \mathcal{P}_{T^\perp}(\boldsymbol{Z}) \right\| \leq \frac{1}{2}.
\tag{12}
$$

Our proof approach is based on the golfing scheme, which was developed and used extensively in prior work Charles et al. (2017); Gross (2011); Candes & Plan (2011); Ahmed et al. (2013). The golphing scheme aims to construct a certificate $\boldsymbol{Z}$ that satisfies the above properties. The golfing scheme is an iterative method that begins with a trivial all-zeros certificate $\boldsymbol{Z}$ and continually refines the certificate through an update iteration that produces a sequence of certificates $\boldsymbol{Z}_k$ for $k \in [1, \cdots, \kappa]$ that each are closer to satisfying the desired conditions. The sequence is then shown to converge to a final certificate $\boldsymbol{Z}_\kappa$ that satisfies both conditions. The golfing iterations are defined as

$$
\boldsymbol{Z}_k = \boldsymbol{Z}_{k-1} + \kappa \mathcal{A}_k^T \mathcal{A}_k (\boldsymbol{Q}\boldsymbol{V}^T - \mathcal{P}_T(\boldsymbol{Z}_{k-1})),
$$

where the operator $\mathcal{A}(\boldsymbol{W})$ represents a single sample of the observation matrix, i.e.,

$$
\mathcal{A}(\boldsymbol{W}) = \text{vec}(\langle \boldsymbol{A}_n, \boldsymbol{W} \rangle).
\tag{13}
$$

CONVERGENCE OF THE GOLFING SCHEME

Using the golfing scheme we need to show that subsequent applications of the iterations converge to a certificate that satisfies the required properties. To show the first property in Equation equation 11 we consider a refined, simpler update iteration used in prior work Ahmed et al. (2013); Charles et al. (2017) deined in terms of a modified certificate $\widetilde{\boldsymbol{Z}}_k = \mathcal{P}_T(\boldsymbol{Z}_k) - \boldsymbol{Q}\boldsymbol{V}^T$:

$$
\widetilde{\boldsymbol{Z}}_k = (\mathcal{P}_T - \kappa \mathcal{P}_T \mathcal{A}_k^T \mathcal{A}_k \mathcal{P}_T)\widetilde{\boldsymbol{Z}}_{k-1},
$$

It follows that one must show that this iterative procedure converges, with high probability, to a certificate satisfying the desired dual certificate conditions. The rest of the details in the appendix are dedicated to this demonstration.

HIGH LEVEL CONVERGENCE OF THE DUAL CERTIFICATE

The first condition we need to demonstrate is that the Forbenious norm of the $k^{th}$ iterate, $\widetilde{\boldsymbol{Z}}_k$ is bounded with high probability.

by using Lemma 4 and observing that the Forbenious norm of the $k^{th}$ iterate is well bounded with probability $1 - O((NT)^{-\beta})$ by

$$
\begin{aligned}
\left\| \widetilde{\boldsymbol{Z}}_k \right\|_F &\leq \max_k \left\| \mathcal{P}_T - \kappa \mathcal{P}_T \mathcal{A}_k^T \mathcal{A}_k \mathcal{P}_T \right\| \left\| \widetilde{\boldsymbol{Z}}_{k-1} \right\|_F \\
&\leq 2^{-k} \left\| \widetilde{\boldsymbol{Z}}_0 \right\|_F \\
&\leq 2^{-k} \left\| \boldsymbol{Q}\boldsymbol{V}^T \right\|_F \\
&\leq 2^{-k}\sqrt{R},
\end{aligned}
$$

Where here we use Lemma 4 so long that $M \geq c\beta\kappa R(\mu_b^2 T + N)\log^2(TN)$. As in Ahmed et al. (2013); Charles et al. (2017) we set $\kappa \geq 0.5\log_2(8\gamma^2 R)$, which reduces this bound for the Frobenious norm of $\widetilde{\boldsymbol{Z}}_\kappa$ is bounded by $\left\| \widetilde{\boldsymbol{Z}}_\kappa \right\|_F \leq (2\sqrt{2}\gamma)^{-1}$.

Next, we show that the second dual certificate holds:

$$
\begin{aligned}
\left\| \mathcal{P}_{T^\perp}\left( \boldsymbol{Z}_\kappa \right) \right\| &\leq \sum_{k=1}^{\kappa} \left\| \mathcal{P}_{T^\perp}\left( \kappa \mathcal{A}_k^T \mathcal{A}_k \widetilde{\boldsymbol{Z}}_{k-1} \right) \right\| \\
&= \sum_{k=1}^{\kappa} \left\| \mathcal{P}_{T^\perp}\left( \kappa \mathcal{A}_k^T \mathcal{A}_k \widetilde{\boldsymbol{Z}}_{k-1} - \widetilde{\boldsymbol{Z}}_{k-1} \right) \right\| \\
&\leq \sum_{k=1}^{\kappa} \left\| \kappa \mathcal{A}_k^T \mathcal{A}_k \widetilde{\boldsymbol{Z}}_{k-1} - \widetilde{\boldsymbol{Z}}_{k-1} \right\| \\
&\leq \sum_{k=1}^{\kappa} \left\| \kappa \mathcal{A}_k^T \mathcal{A}_k \widetilde{\boldsymbol{Y}}_{k-1} - \widetilde{\boldsymbol{Z}}_{k-1} \right\|_F \\
&\leq \sum_{k=1}^{\kappa} \max_{k \in [1,\dots\kappa]} \left\| \kappa \mathcal{A}_k^T \mathcal{A}_k \widetilde{\boldsymbol{Z}}_{k-1} - \widetilde{\boldsymbol{Z}}_{k-1} \right\|_F \\
&\leq \sum_{k=1}^{\kappa} \frac{1}{2}2^{-k} \\
&\leq \frac{1}{2}
\end{aligned}
$$

We use Lemma 5 to bound the maximum spectral norm of $\kappa \mathcal{A}^T{}_k \mathcal{A}_k \widetilde{\boldsymbol{Z}}_{k-1} - \widetilde{\boldsymbol{Z}}_{k-1}$ with probability $1 - O((TN)^{1-\beta})$. Taking $\kappa \geq \log(LN)$ shows that the final certificate $\boldsymbol{Z}_\kappa$ satisfies all the desired properties, completing the proof.

USEFUL DEFINITIONS AND THEOREMS

**Theorem 2** (Matrix Bernstein's Inequality). *Let $\boldsymbol{X}_i \in \mathbb{R}^{L,N}$, $i \in [1,\dots,M]$ be $M$ random matrices such that $\mathbf{E}[\boldsymbol{X}_i] = 0$ and $\|\boldsymbol{X}_i\|_{\psi_\alpha} < U_\alpha < \infty$ for some $\alpha \geq 1$. Then with probability $1 - e^{-t}$, the spectral norm of the sum is bounded by*

$$
\left\| \sum_{i=1}^{M} \boldsymbol{X}_i \right\| \leq C \max\left\{ \sigma_X \sqrt{t + \log(L+N)}, U_\alpha \log^{1/\alpha}\left( \frac{MU_\alpha^2}{\sigma_X^2} \right)(t + \log(L+N)) \right\},
$$

*for some constant $C$ and the variance parameter defined by*

$$
\sigma_X = \max\left\{ \left\| \sum_{i=1}^{M} \mathbf{E}[\boldsymbol{X}_i \boldsymbol{X}_i^*] \right\|^{1/2}, \left\| \sum_{i=1}^{M} \mathbf{E}[\boldsymbol{X}_i^* \boldsymbol{X}_i] \right\|^{1/2} \right\}.
$$

*where Orlicz-$\alpha$ norm $\|X\|_{\psi_\alpha}$ is defined as*

$$
\|X\|_{\psi_\alpha} = \inf\left\{ y > 0 \mid \mathbf{E}\left[ e^{\|X\|^\alpha / y^\alpha} \right] \leq 2 \right\}. \tag{14}
$$

An additional useful lemma from Tropp (2012); Ahmed & Romberg (2014) relates the Orlicz-1 and -2 norms for a random variable and it's square. For completeness, we include this lemma here for easy reference:

**Lemma 1** (Lemma 5.14, Tropp (2012)). *A random variable $X$ is sub-gaussian iff $X^2$ is subexponential. Furthermore,*

$$\|X\|_{\psi_2}^2 \leq \left\|X^2\right\|_{\psi_1} \leq 2\|X\|_{\psi_2}^2 .$$

**Lemma 2** (Lemma 7, Ahmed et al. (2013)). *Let $X_1$ and $X_2$ be two sub-gaussian random variables. Then the product $X_1 X_2$ is a sub-exponential random variable with*

$$\|X_1 X_2\|_{\psi_1} \leq c\|X_1\|_{\psi_2}\|X_2\|_{\psi_2} .$$

## B.1 Bounding the Frobenius norm

**Lemma 3.** *The Frobenius norm of the projection $\mathcal{P}_T\left(\boldsymbol{A}_n\right)$ is bounded by*

$$\left\|\mathcal{P}_T\left(\boldsymbol{A}_n\right)\right\|_F^2 \leq \left\|\boldsymbol{Q}^T\boldsymbol{b}_n\right\|_2^2 + \eta\left\|\boldsymbol{V}^T\boldsymbol{s}_n\right\|_2^2 \tag{15}$$

*where $\boldsymbol{A}_n = \boldsymbol{b}_n\boldsymbol{s}_n^T$ represents rank-one samples and blurring, and $\eta = \|\boldsymbol{b}_n\|_2^2$ is the total fluorescence integration over the PSF.*

*Proof.* Next, we show that $\left\|\mathcal{P}_T\left(\boldsymbol{A}_n\right)\right\|_F^2$, the size of the projection in Frobenius norm is controlled by the number of the samples and blur. Indeed, it follows by algebra that

$$\begin{aligned}
\left\|\mathcal{P}_T\left(\boldsymbol{A}_n\right)\right\|_F^2 &= \left\langle \mathcal{P}_T\left(\boldsymbol{A}_n\right), \mathcal{P}_T\left(\boldsymbol{A}_n\right) \right\rangle \\
&= \left\langle \mathcal{P}_T\left(\boldsymbol{A}_n\right), \boldsymbol{A}_n \right\rangle \\
&= \left\langle \boldsymbol{Q}\boldsymbol{Q}^T\boldsymbol{b}_n\boldsymbol{s}_n^T, \boldsymbol{b}_n\boldsymbol{s}_n^T \right\rangle + \left\langle \boldsymbol{b}_n\boldsymbol{s}_n^T\boldsymbol{V}\boldsymbol{V}^T, \boldsymbol{b}_n\boldsymbol{s}_n^T \right\rangle - \left\langle \boldsymbol{Q}\boldsymbol{Q}^T\boldsymbol{b}_n\boldsymbol{s}_n^T\boldsymbol{V}\boldsymbol{V}^T, \boldsymbol{b}_n\boldsymbol{s}_n^T \right\rangle \\
&= \|\boldsymbol{s}_n\|_2^2\left\|\boldsymbol{Q}^T\boldsymbol{b}_n\right\|_2^2 + \|\boldsymbol{b}_n\|_2^2\left\|\boldsymbol{V}^T\boldsymbol{s}_n\right\|_2^2 - \left\|\boldsymbol{Q}^T\boldsymbol{b}_n\right\|_2^2\left\|\boldsymbol{V}^T\boldsymbol{f}_n\right\|_2^2 \\
&\leq \left\|\boldsymbol{Q}^T\boldsymbol{b}_n\right\|_2^2 + \eta\left\|\boldsymbol{V}^T\boldsymbol{s}_n\right\|_2^2,
\end{aligned} \tag{16}$$

where we use here the fact that $\boldsymbol{A}_n$ for one sample is rank. Therefore,

$$\left\|\mathcal{P}_T\left(\boldsymbol{A}_n\right)\right\|_F^2 \leq \left\|\boldsymbol{Q}^T\boldsymbol{b}_n\right\|_2^2 + \eta\left\|\boldsymbol{V}^T\boldsymbol{s}_n\right\|_2^2$$

$\square$

## B.2 Bounding the Operator Norm

**Lemma 4.** *Suppose $\mathcal{A}_k$ is a measurement operator with $\mathbf{E}\left[\mathcal{A}_k^*\mathcal{A}_k\right] = \frac{1}{\kappa}\mathcal{I}$, where $\boldsymbol{A}_n = \boldsymbol{b}_n\boldsymbol{s}_n^T$ represents rank-one samples. Then with probability at least $1 - \frac{1}{(N+L)}$, we have*

$$\max_{k \in [1, \ldots, \kappa]} \left\|\kappa\mathcal{P}_T\mathcal{A}_k^T\mathcal{A}_k\mathcal{P}_T - \mathcal{P}_T\right\|_{op} \leq \frac{1}{2}$$

*provided that $M \geq C\frac{R}{M}\left(\mu_b^2 T + N\right)\log^2(NT)$ for a large enough constant $C$.*

*Proof.* The goal of this lemma is to prove that with overwhelming probability the operator norm of the function defining the golfing scheme is small, and therefore the certificate norm shrinks by enough to satisfy the dual certificate conditions in $\kappa$ steps. We can achieve the desired bound through a matrix Bernstein inequality. We start by noting that since $\mathbf{E}\left[\mathcal{A}_k^*\mathcal{A}_k\right] = \frac{1}{\kappa}\mathcal{I}$, the operator whose norm we wish to bound is equivalent to

$$\kappa \mathcal{P}_T \mathcal{A}_k^T \mathcal{A}_k \mathcal{P}_T - \mathcal{P}_T = \kappa \mathcal{P}_T \mathcal{A}_k^* \mathcal{A}_k \mathcal{P}_T - \mathbf{E}\left[\kappa \mathcal{P}_T \mathcal{A}_k^T \mathcal{A}_k \mathcal{P}_T\right]$$

$$= \kappa \sum_{n \in \Gamma_k} \left(\mathcal{P}_T\left(\boldsymbol{A}_n\right) \otimes \mathcal{P}_T\left(\boldsymbol{A}_n\right) - \mathbf{E}\left[\mathcal{P}_T\left(\boldsymbol{A}_n\right) \otimes \mathcal{P}_T\left(\boldsymbol{A}_n\right)\right]\right)$$

Next we define the operator $\mathcal{L}_n(\boldsymbol{C}) = \langle \mathcal{P}_T\left(A\right), \boldsymbol{C}\rangle \mathcal{P}_T\left(A\right)$. Note that the Frobenious norm of this operator is $\left\|\mathcal{L}(\boldsymbol{C})\right\|_F^2 = \langle \mathcal{L}(\boldsymbol{C}), \mathcal{L}(\boldsymbol{C})\rangle = \langle \mathcal{L}(\boldsymbol{C}), \mathcal{P}_T\left(\boldsymbol{C}\right)\rangle$ for any $\boldsymbol{C} \in M\left(n, \mathbb{R}\right)$, meaning that we can simplify the above as

$$\kappa \mathcal{P}_T \mathcal{A}_k^T \mathcal{A}_k \mathcal{P}_T - \mathbf{E}\left[\kappa \mathcal{P}_T \mathcal{A}_k^T \mathcal{A}_k \mathcal{P}_T\right] = \kappa \sum_{n \in \Gamma_k} \left(\mathcal{L}_n - \mathbf{E}\left[\mathcal{L}_n\right]\right)$$

**Computing the variance:** To leverage the Bernstein's inequality and bound the operator norm, we need to compute the variance of the random operator. We observe that the projection $\mathcal{P}_T$ is bisectorial and so it suffices to use the symmetry $\mathcal{L}_n$ to calculate

$$\kappa^2 \left\|\sum_{n \in \Gamma_k} \mathbf{E}\left[\mathcal{L}_n^2\right] - \mathbf{E}\left[\mathcal{L}_n\right]^2\right\| \leq \kappa^2 \left\|\sum_{n \in \Gamma_k} \mathbf{E}\left[\mathcal{L}_n^2\right]\right\| = \kappa^2 \left\|\mathbf{E}\left[\sum_{n \in \Gamma_k} \left\|\mathcal{P}_T\left(\boldsymbol{A}_n\right)\right\|_F^2 \mathcal{L}_n\right]\right\|$$

Recall that $\boldsymbol{A}_n = \boldsymbol{b}_n \boldsymbol{s}_n^T \in M\left(\mathbb{R}, 1\right)$ is a rank-1 matrix. Since the point spread function $\boldsymbol{b}_n$ is a standard Gaussian shape and $\boldsymbol{s}_n$ is a sample vector that has at most 1 pixel selected, their respective vector norms are $\|\boldsymbol{b}_n\|_2^2 = \eta$ and $\|\boldsymbol{s}_n\|_2^2 = 1$, where $\eta$ is the total integrated fluorescence power the PSF. We can then use the bound on the Frobenious norm of $\mathcal{P}_T\left(\boldsymbol{A}_n\right)$ from Lemma 3 and substitute to obtain:

$$\kappa^2 \left\|\mathbf{E}\left[\sum_{n \in \Gamma_k} \left\|\mathcal{P}_T\left(\boldsymbol{A}_n\right)\right\|_F^2 \mathcal{L}_n\right]\right\| \leq \kappa^2 \left\|\mathbf{E}\left[\sum_{n \in \Gamma_k} \left(\left\|\boldsymbol{Q}^T \boldsymbol{b}_n\right\|_2^2 + \eta \left\|\boldsymbol{V}^T \boldsymbol{s}_n\right\|_2^2\right) \mathcal{L}_n\right]\right\|$$

$$= \kappa^2 \left\|\mathbf{E}\left[\sum_{n \in \Gamma_k} \left\|\boldsymbol{Q}^T \boldsymbol{b}_n\right\|_2^2 \mathcal{L}_n + \eta \sum_{n \in \Gamma_k} \left\|\boldsymbol{V}^T \boldsymbol{s}_n\right\|_2^2 \mathcal{L}_n\right]\right\|$$

To simplify this bound further, we define the following coherence terms:

$$\mu_b^2 \quad = \quad \frac{N}{R} \max \left\|\boldsymbol{Q}^* \boldsymbol{b}\right\|_2^2. \tag{17}$$

This term quantifies the average intersection of the fluorescing components in the image with the spatial sampling pattern. Specifically, $\mu_b^2$, measures the average intersection between the PSF at any given location with the objects in the image. At a minimum, $\mu_b^2$ being low requires that the PSF does not have the shape of a single cell. Interestingly, due to the correlations between cells and neuropil, the actual overlap is much smaller as the singular vectors of a typical FOV contain widespread information.

Using the definitions of $\mu_b^2$, we can now rewrite the variance term as

$$
\left\| \mathbf{E}\left[ \sum_{n\in\Gamma_k} \left\| \mathcal{P}_T\left(\boldsymbol{A}_n\right)\right\|_F^2 \mathcal{L}_n \right]\right\| \leq \left\| \sum_{n\in\Gamma_k} \mathbf{E}\left[ \left( \left\|\boldsymbol{Q}^T\boldsymbol{b}_n\right\|_2^2 + \eta \left\|\boldsymbol{V}^T\boldsymbol{s}_n\right\|_2^2 \right) \mathcal{L}_n \right]\right\|
$$

$$
\leq \left\| \sum_{n\in\Gamma_k} \mathbf{E}\left[ \left\|\boldsymbol{V}^T\boldsymbol{s}_n\right\|_2^2 \mathcal{L}_n \right]\right\| + \max\left\|\boldsymbol{Q}^T\boldsymbol{b}_n\right\|_2^2 \left\| \sum_{n\in\Gamma_k} \mathbf{E}\left[\mathcal{L}_n\right]\right\|
$$

$$
\leq \frac{R}{T}\left\| \mathbf{E}\left[ \sum_{n\in\Gamma_k} \mathcal{L}_n \right]\right\| + \frac{R}{N}\mu_b^2 \left\| \sum_{n\in\Gamma_k} \mathbf{E}\left[\mathcal{L}_n\right]\right\|
$$

$$
\leq \left( \frac{R}{T} + \frac{R}{N}\mu_b^2 \right)\left\| \sum_{n\in\Gamma_k} \mathbf{E}\left[\mathcal{L}_n\right]\right\|
$$

We thus next focus on $\left\|\sum_{n\in\Gamma_k} \mathbf{E}\left[\mathcal{L}_n\right]\right\|$

$$
\left\| \sum_{n\in\Gamma_k} \mathbf{E}\left[\mathcal{L}_n\right]\right\| = \left\| \sum_{n\in\Gamma_k} \mathbf{E}\left[\mathcal{L}_n\right]\right\| \tag{18}
$$

$$
= \left\| \sum_{n\in\Gamma_k} \mathbf{E}\left[ \mathcal{P}_T\left(\boldsymbol{A}_n\right) \otimes \mathcal{P}_T\left(\boldsymbol{A}_n\right)\right]\right\| \tag{19}
$$

$$
\leq \left\|\mathcal{P}_T\right\| \left\| \sum_{n\in\Gamma_k} \mathbf{E}\left[ \boldsymbol{A}_n \otimes \boldsymbol{A}_n \right]\right\| \left\|\mathcal{P}_T\right\| \tag{20}
$$

$$
= \left\| \sum_{n\in\Gamma_k} \mathbf{E}\left[ \boldsymbol{b}_n\boldsymbol{s}_n \otimes \boldsymbol{b}_n\boldsymbol{s}_n \right]\right\| \tag{21}
$$

$$
\leq \left\| \sum_{n\in\Gamma_k} \mathbf{E}\left[ \{\boldsymbol{b}_n[i]\boldsymbol{b}_n[j]\boldsymbol{s}_n\boldsymbol{s}_n^T\}_{ij} \right]\right\|_F^2 \tag{22}
$$

$$
\leq \frac{1}{T^2}\left\| \sum_{n\in\Gamma_k} \{\mathbf{E}\left[ \boldsymbol{b}_n[i]\boldsymbol{b}_n[j]\right]\boldsymbol{I}\}_{ij} \right\|_F^2 \tag{23}
$$

$$
\leq \frac{T}{T^2 L'^2}\left\| \sum_{n\in\Gamma_k} \boldsymbol{b}_n\boldsymbol{b}_n^T \right\|_F^2 \tag{24}
$$

$$
\leq \frac{T}{M^2}\left\| \sum_{n\in\Gamma_k} \boldsymbol{b}_n\boldsymbol{b}_n^T \right\|_F^2 \tag{25}
$$

$$
\leq \frac{T}{M^2} \sum_{n\in\Gamma_k} \left\| \boldsymbol{b}_n\boldsymbol{b}_n^T \right\|_F^2 \tag{26}
$$

$$
\leq \frac{T}{M^2} \sum_{n\in\Gamma_k} N = \frac{TN}{M^2}\frac{M}{\kappa} = \frac{TN}{M\kappa} \tag{27}
$$

Thus,

$$\sigma_X^2 \leq \left( \frac{R}{T} + \frac{R}{N}\mu_b^2 \right) \left( \frac{NT}{M\kappa} \right) \tag{28}$$

$$= \frac{R\left(N + T\mu_b^2\right)}{M\kappa} \tag{29}$$

**Computing the Orlicz norm:** The second term of the Bernsteins inequality requires the Orlicz-1 norm. Since $\mathcal{L}_n$ is positive semidefinite (PSD), it's a Young function and thus we can compute its Orlicz-1 norm:

$$\kappa \Big\| \|\mathcal{L}_n\|_F - \mathbf{E}\left[\|\mathcal{L}_n\|_F\right] \Big\|_{\psi_1}$$

We would like to show that with high probability that the projection operator of measurements is small with high probability. In order to do this , we seek to first compute the size of the operator by measuring it in the Frobenius sense and then we bound it using the Matrix Bernstein inequality.

$$\Big\| \|\mathcal{L}_n\|_F - \mathbf{E}\left[\|\mathcal{L}_n\|_F\right] \Big\|_{\psi_1} \leq \max \left\{ \Big\| \|\mathcal{L}_n\|_F \Big\|_{\psi_1} - \Big\| \mathbf{E}\left[\|\mathcal{L}_n\|_F\right] \Big\|_{\psi_1} \right\}$$

First, note that the size of $\Big\| \mathbf{E}\left[\|\mathcal{L}_n\|_F\right] \Big\|_{\psi_1}$ can be determined via

$$\Big\| \mathbf{E}\left[\|\mathcal{L}_n\|_F\right] \Big\|_{\psi_1} = \Big\| \mathbf{E}\left[ \mathrm{Tr}\, \mathcal{P}_T \mathcal{A}_n \mathcal{A}_n^T \mathcal{P}_T^T \right] \Big\|_{\psi_1}$$

$$= \Big\| \mathbf{E}\left[ \mathrm{Tr}\, \mathcal{A}_n \mathcal{I} \mathcal{A}_n^T \right] \Big\|_{\psi_1} \tag{30}$$

$$= \frac{\eta}{T \log(2)}$$

Next, to compute $\Big\| \|\mathcal{L}_n\|_F \Big\|_{\psi_1}$ we use the definition of the Orlicz-1 norm in Equation equation 14 to see that

$$\Big\| \|\mathcal{L}_n\|_F^2 \Big\|_{\psi_1} = \inf \left\{ K > 0 : \mathbf{E}\left[ \exp\left( \|\mathcal{L}_n\|_F^2 / K \right) \leq 2 \right] \right\} \tag{31}$$

$$= \inf \left\{ K > 0 : \exp\left( \left\| \mathcal{P}_T\left( \boldsymbol{A}_n \right) \right\|_2^2 / K \right) \leq 2 \right\} \tag{32}$$

$$\leq \inf \left\{ K > 0 : \exp\left( \left\| \boldsymbol{Q}^T \boldsymbol{b}_n \right\|_2^2 + \eta \left\| \boldsymbol{V}^T \boldsymbol{s}_n \right\|_2^2 / K \right) \leq 2 \right\} \tag{33}$$

$$\leq \inf \left\{ K > 0 : \exp\left( R\mu_b^2/N + 1/T/K \right) \leq 2 \right\} \tag{34}$$

$$\tag{35}$$

using the fact that the norm is non-negative, the coherence term and definition of $\boldsymbol{s}_n$. It follows that for $K$ large enough that

$$\Big\| \|\mathcal{L}_n\|_F^2 \Big\|_{\psi_1} = \frac{TR\mu_b^2 + N}{NT \log(2)} \leq R\frac{T\mu_b^2 + N}{M} \tag{36}$$

We now have bounds on the variance and the Orliczs norm and thus can provide a bound on the largest singular value of the operator via Theorem 2. Specifically, we can see that the first term in Theorem 2 using $t = \beta \log(TN) > \log(N + T)$ for some $\beta > 0$ and with $C$ large enough

$$\sigma_X \sqrt{t + \log\left(N + T\right)} \leq \sqrt{2\beta R \frac{(N + T\mu_b^2)}{M\kappa} \log\left(NT\right)}. \tag{37}$$

Similarly, we can bound the second term

$$U_1 \log\left(\frac{MU_1^2}{\sigma_X^2}\right)\left(t + \log\left(N + T\right)\right) \leq 2\beta U_1 \log\left(\kappa M\right) \log\left(NT\right) \tag{38}$$

$$\leq 2\beta R \frac{(N + T\mu_b^2)}{M} \log\left(cR(N + T\mu_b^2)\right) \log\left(NT\right) \tag{39}$$

$$\leq 2\beta R \frac{(N + T\mu_b^2)}{M} \log^2\left(NT\right) \tag{40}$$

Putting these two terms together in the Bernstein's inequality yields

$$\left\|\kappa \mathcal{P}_T \mathcal{A}_k^T \mathcal{A}_k \mathcal{P}_T - \mathcal{P}_T\right\| \leq c \max\left[R\frac{(N + T\mu_b^2)}{M} \log^2\left(NT\right), \sqrt{2\beta R \frac{(N + T\mu_b^2)}{M\kappa} \log\left(NT\right)}\right] \tag{41}$$

Now, by assuming that

$$M \geq C\frac{R}{M}\left(\mu_b^2 T + N\right) \log^2(NT) \tag{42}$$

and take the union bound over the $\kappa$ partitions, we complete the proof.

$\square$

## B.3 BOUNDING CERTIFICATES

**Lemma 5.** *Let $\mathcal{A}_k$ be defined as in Equation equation 13, $\kappa < M$ be the number of steps in the golfing scheme and assume that $M \leq TN$. Then as long as*

$$M \geq C\kappa R \max\{\mu_b T, N\} \log(NT) \tag{43}$$

*then with probability at least $1 - O(M(TN)^{-\beta})$, we have*

$$\max_k \left\|\kappa \mathcal{A}_k^T \mathcal{A}_k(\widetilde{\boldsymbol{Z}}_{k-1}) - \widetilde{\boldsymbol{Z}}_{k-1}\right\| \leq 2^{-(k+1)}.$$

*Proof.* The proof is based on bounding the operator norm of $\kappa \mathcal{A}^T \mathcal{A} - \mathcal{I}$. As with the previous lemma, the core of this proof is the use of the matrix Bernstein Inequality 2. Consider an arbitrary matrix $\boldsymbol{G}$ such that,

$$X_n = \kappa(\langle \boldsymbol{G}, \boldsymbol{A}_n\rangle \boldsymbol{A}_n - \mathbf{E}\left[\langle \boldsymbol{G}, \boldsymbol{A}_n\rangle \boldsymbol{A}_n\right]). \tag{44}$$

Our goal is to first control $\left\| \sum \mathbf{E}\left[X_n X_n^T\right] \right\|$ and $\left\| \sum \mathbf{E}\left[X_n^T X_n\right] \right\|$. We start with the latter of these terms, $\left\| \sum \mathbf{E}\left[X_n^T X_n\right] \right\|$:

$$
\begin{aligned}
\left\| \sum_{n \in \Gamma_k} \mathbf{E}\left[X_n^T X_n\right] \right\| &\leq \kappa^2 \left\| \sum_{n \in \Gamma_k} \mathbf{E}\left[|\langle \boldsymbol{G}, \boldsymbol{A}_n \rangle|^2 \boldsymbol{A}_n \boldsymbol{A}_n^* \right] \right\| \\
&\leq \kappa^2 \left\| \sum_{n \in \Gamma_k} \mathbf{E}\left[\|\boldsymbol{G}\boldsymbol{b}\|_2^2 \|\boldsymbol{s}_n^T\|_2^2 \boldsymbol{b}_n \boldsymbol{s}_n^T \boldsymbol{s}_n \boldsymbol{b}_n^T \right] \right\| \\
&\leq \frac{\kappa^2}{T} \max \|\boldsymbol{G}\boldsymbol{b}\|_2^2 \left\| \sum_{n \in \Gamma_k} \mathbf{E}\left[\boldsymbol{b}_n \boldsymbol{b}_n^T \right] \right\| \\
&= \frac{\mu_b T \kappa^2}{L'^2 T^2} \|\boldsymbol{G}\|_F^2 \left\| \sum_{n \in \Gamma_k} \boldsymbol{I} \right\| \\
&\leq \frac{\mu_b T \kappa}{M} \|\boldsymbol{G}\|_F^2 \tag{45}
\end{aligned}
$$

For the first term, $\left\| \sum \mathbf{E}\left[X_n X_n^T\right] \right\|$, we can similarly bound

$$
\begin{aligned}
\left\| \sum_{n \in \Gamma_k} \mathbf{E}\left[X_n X_n^T\right] \right\| &\leq \kappa^2 \left\| \sum_{n \in \Gamma_k} \mathbf{E}\left[|\langle \boldsymbol{G}, \boldsymbol{A}_n \rangle|^2 \boldsymbol{A}_n^T \boldsymbol{A}_n \right] \right\| \\
&= \frac{N \kappa^2}{L'^2} \left\| \sum_{n \in \Gamma_k} \mathbf{E}\left[\|\boldsymbol{G}\boldsymbol{s}_n\|_2^2 \boldsymbol{s}_n \boldsymbol{s}_n^T \right] \right\| \\
&= \frac{N \kappa^2}{L'^2 T^2} \left\| \sum_{n \in \Gamma_k} \mathbf{E}\left[\|\boldsymbol{G}\boldsymbol{s}_n\|_2^2 \right] \right\| \\
&\leq \frac{N \kappa^2}{L'^2 T^2} \|\boldsymbol{G}\|_F^2 \left\| \sum_{n \in \Gamma_k} \boldsymbol{I} \right\| \\
&\leq \frac{N \kappa}{M} \|\boldsymbol{G}\|_F^2 \tag{46}
\end{aligned}
$$

Thus, we can write that

$$
\sigma_X^2 \leq \frac{\kappa}{M} \|\boldsymbol{G}\|_F^2 \max\{\mu_b^2 T, N\} \tag{47}
$$

To use the Bernstein's inequality, as with Lemma 4, we need to bound the Orlicz norm for $\alpha = 1$.

$$U_1 = \|X\|_{\psi_1} \tag{48}$$

$$\leq 2\kappa\left\|\left\|\langle \boldsymbol{G}, \boldsymbol{A}_n\rangle \boldsymbol{A}_n\right\|_F^2\right\|_{\psi_1} \tag{49}$$

$$\leq c\kappa\left\|\langle \boldsymbol{G}, \boldsymbol{A}_n\rangle\right\|_{\psi_2}\left\|\|\boldsymbol{A}_n\|_F\right\|_{\psi_2} \tag{50}$$

$$\leq c\kappa\left\|\mathrm{Tr}\, \boldsymbol{s}_n\boldsymbol{b}_n^T\boldsymbol{G}\right\|_{\psi_2}\sqrt{\left\|\|\boldsymbol{b}_n\|_2^2\|\boldsymbol{s}_n\|_2^2\right\|_{\psi_2}} \tag{51}$$

$$\leq c\kappa\left\|\boldsymbol{b}_n^T\boldsymbol{G}\boldsymbol{s}_n\right\|_{\psi_2} \tag{52}$$

$$\leq c\kappa\left\|\left\|\boldsymbol{G}^T\boldsymbol{b}_n\right\|_2\|\boldsymbol{s}_n\|_2\right\|_{\psi_2} \tag{53}$$

$$\leq c\kappa\left\|\left\|\boldsymbol{G}^T\boldsymbol{b}_n\right\|_2\right\|_{\psi_2} \tag{54}$$

$$\leq c\kappa\sqrt{\frac{1}{L'}}\|\boldsymbol{G}\|_F \tag{55}$$

$$\leq c\kappa\sqrt{\frac{T}{M}}\|\boldsymbol{G}\|_F \tag{56}$$

We can now apply the matrix Bernstein theorem with the calculated values of $U_1$ and $\sigma_X$. Again, using $t = \log(LN) > \log(N+L)$, the bound is

$$\sigma_X\sqrt{t + \log(T+N)} \leq \kappa\sqrt{\beta\frac{\kappa}{M}\|\boldsymbol{G}\|_F^2\max\{\mu_b T, N\}\log(TN)} \tag{57}$$

$$\leq \kappa c 2^{-k}\sqrt{\beta\frac{\kappa}{M}R\max\{\mu_b T, N\}\log(TN)}. \tag{58}$$

where here we use the bound $\|\boldsymbol{G}\|_F \leq 2^{-k}\sqrt{R}$. Similarly,

$$* = U_1\log\left(\frac{\Delta U_1^2}{\sigma_X^2}\right)(t + \log(N+T)) \tag{59}$$

$$\leq c\kappa\beta\|\boldsymbol{G}\|_F\sqrt{\frac{T\mu_b^2}{M}}\log\left(c^2\kappa^2\frac{M}{\kappa}\frac{MT\|\boldsymbol{G}\|_F^2}{M\kappa R\max(\mu_b^2 T, N)\|\boldsymbol{G}\|_F^2}\right)\log(NT) \tag{60}$$

$$\leq c\kappa\beta\|\boldsymbol{G}\|_F\sqrt{\frac{T}{M}}\log\left(c^2\frac{TM}{\max(\mu_b^2 T, N)}\right)\log(NT) \tag{61}$$

$$\leq c\kappa\beta 2^{-k}\sqrt{\frac{TR}{M}}\log^2(NT) \tag{62}$$

where the last inequality uses the bound $\|\boldsymbol{G}\|_F \leq 2^{-k}\sqrt{R}$.

We can now bound $\mu_k^2 \leq \mu_b^2$ with probability $1 - O(M(LN)^{-\beta})$ and the prior Lemma to bound $\|\boldsymbol{G}_k\|_F \leq 2^{-k}\sqrt{R}$, which gives us a final bound of

$$\left\|(\mathcal{A}^*\mathcal{A} - \mathcal{I})\boldsymbol{G}_k\right\| \leq 2^{-k/2}\max\left[\sqrt{\frac{\kappa\beta R}{M}\max\{\mu_b T, N\}\log(LN)}, c\kappa\beta\sqrt{\frac{RT}{M}}\log^2(NT)\right]. \tag{63}$$

Taking

$$M \geq C\kappa R\max\{\mu_b T, N\}\log(NT) \tag{64}$$

for $C$ large enough and $\kappa = 1$ completes the proof.

□

