# OpenReview forum: "Fast Two-photon Microscopy by Neuroimaging with Oblong Random Acquisition (NORA)"
_ICLR.cc/2026/Conference — Submitted to ICLR 2026_

### Official Review · Reviewer_SyYH · 2025-10-29

**Soundness:** 3
**Presentation:** 3
**Contribution:** 2
**Rating:** 4
**Confidence:** 4

**Summary:**

The paper presents a reconstruction framework for acquisition using a two-photon microscope. After an analysis of the image formation, an optimization problem is build for the reconstruction. The problem is solved using a fast conical optimization scheme. The experiments show some interesting results and the system seems promising for imaging.

**Strengths:**

The reconstruction problem is cast as a matrix completion problem. The following optimization problem is motivated by well know reconstruction analysis and results. In the proposed setting, the main contribution of the paper is the central theorem that gives some insight on the reconstruction error. It gives ideas on how many measurements are needed for an acceptable reconstruction.

**Weaknesses:**

I see several weaknesses in the paper.

- The implementation is an optical framework. Such part will only interest people with instrumentation knowledge. Thus, the ICLR community may be not the best for such contribution.
- There is no analysis of the noise from a physics point of view. What is the nature of the error matrix? Is it random? Is it compound of several error terms?
- Please check how the references are inserted into the LaTeX file and use \citep for paper citation and \citet for title...

**Questions:**

- Does the noise mostly Gaussian or Poissonian? If Poissonian would it be more interesting to take it into account using Anscombe transform (see [1])?
- What is $A_n$ in equation (13)?
- What is the adjoint operator of $\mathcal{A}_k$?


 [1] Azzari, L., & Foi, A. (2016). Variance stabilization for noisy+ estimate combination in iterative Poisson denoising. IEEE signal processing letters, 23(8), 1086-1090.

---

> ### Author Response · Authors · 2025-11-19
> **Responses/clarifcations**
>
> We appreciate the reviewer's time in reading and providing feedback on our manuscript. We would like to take the opportunity to respond:
>
> Weaknesses:
>
> Relevance to ICLR: We agree that there is an application-bend to our manuscript. We feel that the theory we developed, as well as the growing area of functional computational imaging, will be of interest to the ICLR community. For one, our theory focuses on the use of low rank representations for inverse problems. This theory, which is often missing in many more applied settings, provides the justification for such a microscope design, and we feel that the topics of theoretical guarantees on representation-based inverse problems would be of interest to ICLR. Second, the area of computational optics in functional imaging is growing and can intersect richly with the representation learning community. Learning new models for neural imaging data, for example, can continue to improve the capabilities of scientists to learn about the brain. Our work represents one step in this direction, covering theory and practice, and we feel will interest others working on more generic image representations to similarly focus on this area.
>
> Noise modeling: From a theoretical angle, we note that our theory only assumes bounded noise, and that the noise term shows up in our reconstruction bound. Practically, the noise in multi-photon imaging is a combination of the Poisson photon emissions and the Poisson-Gaussian combined PMT noise. This noise is modeled in the NAOMi biophysical simulation suite that we used for validation, and so is captured by our empirical results (i.e., we can recover even in non-Gaussian settings). We have considered implementing other distributions (e.g., Poisson as in [R1] below) as well as the Anscombe transform, however Poisson likelihoods tended to be slow, and the matrix completion setting performed well as is (and matches the theory).
>
> [R1] Charles et al. Stochastic filtering of two-photon imaging using reweighted \ell_1. ICASSP. 2017.
>
> Citations: Thank you for noticing the inconsistancy and we will look over carefully to ensure proper citation is used.
>
> Qustions:
>
> Noise considerations: As per above, the noise in multi-photon imaging is a combination of the Poisson photon emissions and the Poisson-Gaussian combined PMT noise. At high enough rates, as used here, Gaussian is sufficient, as evidenced by 1) the empirical results given that the full distribution is simulated in the NAOMi simulations, and 2) and our theory actually covers any bounded noise. In very dim imaging (e.g., if novel indicators with very low photon emission were used) it might be the case that data becomes more non-Gaussian, with potential gains in using the Anscombe transform or changing the likelihood. However modern neural imaging has very bright indicators (relatively) and so we stayed with the optimization that is covered by our theoretical results and that works on realistic simulations.
>
> Definition of $\textbf{A}_n$: Thank you for noticing this oversight. $\textbf{A}_n$ is defined after Equation (15) as $\textbf{A}_n = \textbf{b}_n\textbf{s}_n^T$: i.e., an outer product representing a single "measurement" of a blur and sample at one location in the spatial field of view (b) and at one time-point (s). We will move up this definition to where An is first used to make sure Equation (13) is fully defined.
>
> Adjoint of $\mathcal{A}_k$: As $\mathcal{A}_k$ represents blur and subsample of the full video, the adjoint is effective the "transpose"; in this case placing the values subsampled within a large zeros array of the size of the video, and then blurring (as convolution is the adjoint of convolution) with the PSF. As the PSF is symmetric, we do not have to flip it to make the adjoint. We will clarify this in the Proof above Equation (13) where we first use the adjoint of $\mathcal{A}_k$.

---

### Official Review · Reviewer_haBw · 2025-10-31

**Soundness:** 3
**Presentation:** 3
**Contribution:** 3
**Rating:** 4
**Confidence:** 5

**Summary:**

The paper proposes a computational two-photon imaging scheme that addresses the speed limit of traditional point-by-point scanning 2P imaging. It skips most raster lines per frame while slightly blurring along the slow-scan axis with an oblong PSF, and reconstructs the full pixels-by-time matrix with nuclear-norm minimization. The theory proves that under a low-rank assumption on the sample, accurate video reconstruction can be achieved under a very low sampling rate. Simulations with NAOMi indicate accurate recovery at 10–20× line-subsampling under realistic noise and motion, preserving ROI time traces.

**Strengths:**

The paper is clearly written with theoretical analysis.

**Weaknesses:**

No experimental demonstration conducted, no adequate comparison with existing techniques.

**Questions:**

1. The simulation of blurring and subsampling (line 313-315) does not follow the real experiment scenario. To simulate the real 2P imaging in your schematic, you should apply subsampling on the ground truth (sample under observation) first, and then apply the blurring PSF. Otherwise, the crosstalk from neighboring (unsampled) lines will mix up with your sampled ones. The forward model, reconstruction algorithm, as well as the theory proofs should be modified accordingly as well.
2. What are the spatial extents of line-by-line and rigid motions being introduced compared to the neuron sizes? It would be great to see the trend in reconstruction quality as motion increases, and under what kinds of motion the reconstruction finally breaks.
3. How does this compare to line-scanning-based 2P imaging, which only needs one galvo scanner and will be, in principle, much faster than this technique?

[1] Tal, Eran, Dan Oron, and Yaron Silberberg. "Improved depth resolution in video-rate line-scanning multiphoton microscopy using temporal focusing." Optics letters 30.13 (2005): 1686-1688.

[2] Xue, Yi, et al. "Scattering reduction by structured light illumination in line-scanning temporal focusing microscopy." Biomedical optics express 9.11 (2018): 5654-5666.

4. What is the exact sensor noise level being introduced in the simulation? How does the reconstruction react to the scattering introduced noise in neurons?

---

> ### Author Response · Authors · 2025-11-19
> **Response/clarifications**
>
> We thank the reviewer for their time and effort in reviewing our manuscript.
>
> We would like to clarify a few key points:
>
> 1. Blurring and subsampling. We would like to clarify that the compression is performed with the optics at the sample. Therefore we are not collecting data that we then subsample/blur, but in fact are sub-sampling points collected with a point-spread function (PSF) that has a non-trivial spread. The PSF illumination spread is the effective blur, which in essense accumulates fluorescence information across a wider area than a typical diffraction-limited PSF. The illumination with the PSF comes before the sub-sampling that is implemented by only illuminating portions (random lines) across the field-of-view. Therefore, the blur-then-subsample does correctly reflect the acutal imaging process and is in line with the theory and simulations. In fact this is exactly how the NAOMi simulation was set up: the PSF was designed to be an elongated PSF (i.e., the blur), and then random lines were sampled by applying that PSF (blur) at a sampling of lines.
>
> 2. Motion: We simulated typical motion as quantified by prior researchers (in particular Collman 2010). This motion is on the order of a few microns (plus-or-minus 10 microns) which is on the order of about half the diameter of a neuron. This motion is semi-random and correlated in both axial directions, as is implemented in the NAOMi simulation which we used to validate our approach. We agree that it could be interesting to test the limits of our imaging approach, however we focused this work on demonstrating that reconstruction succeeds under typical/realistic scenarios.
>
> 3. Relationship to Temporal focusing: Interestingly temporal focusing is similar in nature to NORA. However, in temporal focusing, the raster scanning must cover the full field of view (which fundamentally limits the possible speed-ups) and the blurring is not compensated for in post-processing, which could yield poor neuronal segmentation in practice. NORA avoids both these complications by recovering the full-resolution data and not needing to fully sample every frame. We agree that this is an important distinction on our approach and we will add additional discussion points comparing to temporal focusing and referencing that body of work.
>
> 4. The noise in the simulation is a combination of the Poisson photon emissions and the Poisson-Gaussian combined PMT noise. Interestingly the PMT acts as a sequence of Poisson-Gaussian random variables that can be modeled as a Gaussian whos mean and vairance are driven by the Poisson photon rate. This is all contained in the NAOMi paper and is used when simulating data for our empirical results. We have indeed considered Poisson and other priors, as per our theory we only need the noise to be bounded for the matrix completion recovery to succeed. We note that the scattering is in part compensated for by the background illumination that is also recovered as part of the low-rank solution. Thus the recovery still needs to undergo neuron segmentation (this is a problem addressed in many other papers) including separating neuronal signals from background illumination.

---

### Official Review · Reviewer_GYmL · 2025-11-01

**Soundness:** 3
**Presentation:** 3
**Contribution:** 2
**Rating:** 4
**Confidence:** 5

**Summary:**

This paper proposes NORA to enhance the resolution of standard two-photon microscopy by combining optical blurring, subsampling, and computational reconstruction. Two cylindrical lenses generate an elongated PSF, enabling a blur-and-subsample strategy in which each high-resolution pixel contributes to multiple overlapping wide-line scans. The forward imaging model represents the optical path as a linear blur operator applied to a low-rank fluorescence video matrix, followed by subsampling. Reconstruction is performed via nuclear norm-regularized least-squares optimization, which leverages correlations across frames and the low-rank prior to recover high-resolution pixel values from undersampled measurements.

**Strengths:**

1. NORA achieves accelerated two-photon microscopy acquisition by combining random line scanning with an elongated PSF, which can be implemented with minimal hardware changes.
2. The paper is well-structured, clear, and easy to follow.

**Weaknesses:**

1. The reconstruction in NORA relies on nuclear norm optimization, modeling the fluorescence video as a low-rank matrix. This low-rank prior may not hold when the imaged activity is highly complex or nonlinear, such as in large-scale rapid neural dynamics or highly dynamic cellular structures, potentially leading to degraded reconstruction performance. In other words, if the intrinsic dimensionality of the video significantly exceeds the assumed rank, increasing the number of samples may still be insufficient to accurately recover fine details.
2. Elongating the PSF allows integration of information along the slow-scan direction, reducing the number of required scans; however, it introduces local blurring. While the spatial resolution along the fast-scan axis is preserved, resolution along the slow-scan axis is inevitably compromised, which may limit the method’s suitability for experiments requiring fine structural analysis. Moreover, the extent of PSF elongation is inherently limited and cannot fully cover missing scan lines, making reconstruction still dependent on multi-frame information and the low-rank prior.
3. While NORA leverages low-rank priors and linear forward modeling to achieve high-speed imaging, its performance is inherently limited by assumptions such as low-rank structure and the partial blurring along the slow-scan axis. These limitations raise the question of whether deep learning-based approaches could further enhance imaging performance.
4. The experimental evaluation of NORA is limited by the lack of testing under realistic imaging conditions. Additionally, the study provides limited quantitative analysis of image quality, such as metrics for spatial resolution, reconstruction accuracy, or signal-to-noise ratio, making it difficult to rigorously compare the method against existing approaches or to fully characterize its practical performance.

**Questions:**

Please see the Weaknesses.

---

> ### Author Response · Authors · 2025-11-19
> **Clarifications/response**
>
> We thank the reviewer for their comments and would like to take the opportunity to respond.
>
> Low-rank assumptions: We agree that the low-rank assumption is an assumption that when violated would reduce the quality of reconstruction. We felt that this was an appropriate prior to assume over the data as it is a fundamental prior already in widespread use in processing microscopy data of neural activity (see for exameple citations in our manuscript for both denoising and segmentation of calcium imaging data). We were of course concerned that other aspects of imaging, such as motion, would reduce efficacy, and our simulated experiments using the biophysical simulation suite "NAOMi" provided empirical demosntrations that even motion, which increases the effective rank of the video data, does not hinder accurate reconstruction. We further note that high levels of neural activity (as long as the neurons are independent of each other) can increase the rank as well, however each neuron spans many pixels, meanting that the overall gain in rank for the video is constrained by the spatial coverage. Each pixel cannot, biologically, be independent of all others, serving as further justification for the low-rank assumption.
>
> Elongated PSF: We agree that the PSF elongation does combine information across lines, and this is a purpousful design choice. By subsampling we would lose much of the information on each frame, and the elongated PSF mitigates this loss by collecting that information in aggregate. That said, the reconstruction algortihm inverts this effective blurring, as evidenced by both our theoretical proofs and empirical tests. We can recover (see Figures 3 and 4) data that is not blurred to the same extent as the PSF elongation.
>
> Deep Learning: Deep learning approaches might be able to help improve reconstruction, and this is an interesting point for future exploration that we have considered. In this work we aimed to deomonstrate that even with simple matrix completion using widely validated priors of neural imaging data that such a scheme can work in both theory and practice.
>
> Testing under realistic conditions: The NAOMi simulation suite is precisely this. It is a biophysical simulation that builds in-silico models of neurons, vasuclature, etc. and simulates realistic data with the appropriate noise characteristics, and includting motion and pixel-by-pixel bleedthrough. This simulation has been used by dozens of groups to validate both algorithmic and optical designs, and so we consider this the best version of realistic data that has ground truth for comparison.

---

### Official Review · Reviewer_7SsV · 2025-11-03

**Soundness:** 4
**Presentation:** 3
**Contribution:** 2
**Rating:** 2
**Confidence:** 3

**Summary:**

The authors develop a method for recovering high-resolution neuroimaging videos  from low-resolution measurements (specifically, blurred randomized line sampling).  The method solves  sparse inverse problem, exploiting the spatio-temporal redundancy of the globally moving signal, and is tested on a two-photon microscopy simulator.

**Strengths:**

An important practical problem, and the presentation is generally clear.

**Weaknesses:**

Novelty?  I assume these videos are undergoing globally rigid motion, which induces substantial spatio-temporal redundancy.  I don't know the multi-photon imaging literature well, but this is a well-studied problem in photographic video processing, and many algorithms exist to perform motion-compensated estimation or restoration (including those that underlie video coding systems like MPEG).  This also shows up in the visual neuroscience literature, where some authors have explored how human vision can maintain high acuity when the eys are constantly moving (both large saccadic eye movements, and small fluctuations).  There are currently no citations to any of this literature.

A secondary concern, perhaps more for the area chairs to decide, is whether ICLR is the right venue for this paper.  Although estimation is a theme in the meeting, this paper does not discuss representations, or learning.

**Questions:**

Would be interesting to see a comparison of the compressed-sensing style solution used in the paper against a more traditional translational motion solution,  which assumes the spatio-temporal signal is two dimensional (or, equivalently, that it's Fourier spectrum lies on a plane).  For non-translational (but still smooth) motions, this can be done locally, as is found in many optic flow estimation algorithms.

---

> ### Author Response · Authors · 2025-11-19
> **Clarifications**
>
> We appreciate the time and effort of the reviewer. We would like to clarify what we think might be a misconception of our work. Motion correction is indeed an issue generally in multi-photon imaging, however in this case it is not a fundamental barrier nor a main point of the algorithmic and imaging design or theory. All we aim to show is that theoretically low-rank matrix completion can be coupled with novel optical imaging in order to more quickly record neural data, providing access to faster time-scales in challenging imaging conditions.
>
> In regards to MPEG and human vision acuity, these are indeed interesting topics, however we feel that these are tangential to the core microscopy design that we developed. Our work falls into the compressive data acquisition (unlike video compression post-hoc, such as MPEG) using optical designs. We are also unsure what "motion solutions" are indicated by the reviewer, and would be happy to know of other methods that might leverage our thoeretical contributions to improve practical signal recovery.
>
> With regards to relevance to ICLR, the theory we developed focuses on the use of low rank representations for inverse problems. This theory, which is often missing in many more applied settings, provides the justification for such a microscope design, and we feel that the topics of theoretical guarantees on representaiton-based inverse problems would be of interest to ICLR.

---

### Meta-Review · Area_Chair_c46L · 2026-01-02

**Summary:**

The authors present a sampling strategy and low-rank matrix recovery method for two-photon microscopy that promises the ability to scan faster with fewer samples and do faithful reconstruction. Most reviewers raised issues related to (1) the appropriateness of the work for ICLR (2) the limitations inherent in a low-rank matrix assumption (namely that the sample does not move too much). I should note that there is already significant work in the two-photon world addressing the issue of registration in a moving sample (e.g. Suite2p, Pachitariu et al (2017), https://www.biorxiv.org/content/10.1101/061507v2.abstract) which this work does not even touch on. I concur with the reviewers that the paper is not at the level to be accepted at ICLR. I would recommend that a future version which addresses reviewer concerns more thoroughly and properly engages with the past work in this area would go to a methods journal instead.

**Reviewer Concerns:**

The reviewers addressed many of the specific concerns, but the broader questions about appropriateness for ICLR and issues with registration which previous (uncited) work had already addressed were of too large a scope to be addressed in the rebuttals.

**Reviewer Scores:**

It's hard to say. Perhaps some of the reviewers who gave a 4 would have bumped it up to a 5, but not enough to change the overall decision.

---

### Decision · Program_Chairs · 2026-01-26

Reject